# A Probiotic Bacterium with Activity against the Most Frequent Bacteria and Viruses Causing Pediatric Diarrhea: *Bifidobacterium longum* subsp. *infantis* CECT 7210 (*B. infantis* IM1^®^)

**DOI:** 10.3390/microorganisms12061183

**Published:** 2024-06-11

**Authors:** José Antonio Moreno-Muñoz, Jesús Delgado Ojeda, Jesús Jiménez López

**Affiliations:** Laboratorios Ordesa S.L., Parc Científic de Barcelona, C/Baldiri Reixac 15-21, 08028 Barcelona, Spain; jesusojeda82@gmail.com (J.D.O.); jesus.jimenez@ordesalab.com (J.J.L.)

**Keywords:** *Bifidobacterium*, probiotic, microbiota, pathogen, gut

## Abstract

The second leading cause of death in children under five years old is diarrheal disease. Probiotics, specifically bifidobacteria, have been associated with a reduction in the number of diarrhea episodes and their severity in babies. In this paper, we summarize the preclinical and clinical evidence of the efficacy of *B. longum* subsp. *infantis* IM1^®^ against various gastrointestinal pathogens using in vitro models, animal models, and clinical studies carried out in our laboratory. The preclinical data demonstrate that IM1^®^ effectively inhibits rotavirus replication (by up to 36.05%) in MA-104 and HT-29 cells and from infection (up to 48.50%) through the production of an 11-amino-acid peptide. IM1^®^ displays the capability to displace pathogens from enterocytes, particularly *Cronobacter sakazakii* and *Salmonella enterica*, and to reduce the adhesion to the HT29 cells of *C. sakazakii* and *Shigella sonnei*. In animal models, the IM1^®^ strain exhibits in vivo protection against rotavirus and improves the clinical symptomatology of bacterial gastroenteritis. A clinical study involving infants under 3 months of age revealed that IM1^®^ reduced episodes of diarrhea, proving to be safe, well tolerated, and associated with a lower prevalence of constipation. *B. infantis* IM1^®^ emerges as an effective probiotic, diminishing episodes of diarrhea caused by gastrointestinal pathogens.

## 1. Introduction

Acute gastrointestinal infections are the main cause of gastroenteritis among children around the world. Enteroviruses such as rotavirus, norovirus, and astrovirus and bacterial pathogens such as *Shigella*, *Salmonella*, *Escherichia*, and *Campylobacter*, among others, are the main causative agents of gastroenteritis.

Gastroenteritis contributes to approximately 10 percent of pediatric deaths and stands as the second leading cause of death globally. Among infants under 24 months of age, rotavirus stands out as the primary cause, while *Shigella* takes precedence as the most common cause after the age 24 months, with rotavirus following closely as the second most prevalent.

While the occurrence of rotavirus infection is comparable between high-income and low–middle-income countries, a staggering 80% of deaths take place in developing nations. In 2013, approximately half (49%) of all estimated rotavirus deaths were concentrated in four countries—India, Nigeria, Pakistan, and the Democratic Republic of Congo.

The fecal–oral route is how rotavirus is transmitted, with the infectious dose being 100 virus particles [1]. Rotavirus infects fully developed enterocytes of the intestinal villus, sparing crypt cells [2]. Upon infiltration into epithelial cells, it produces the NSP4 enterotoxin, initiating phospho-lipase C-dependent cell signaling and increasing intracellular calcium levels, producing chloride secretion [3]. Rotavirus-induced diarrhea is ascribed to diverse mechanisms, encompassing secondary malabsorption, the entero-toxic role of NSP4, the destruction of enterocytes, the stimulation of the enteric nervous system, and villus ischemia [4]. The consequence is abundant watery diarrhea lasting 2 to 7 days, inducing fluid and electrolyte loss that might result in fatal dehydration. Timely and vigorous rehydration with oral or intravenous fluids can rectify these imbalances and sustain a child until the diarrhea subsides [5]. Moreover, viremia has been documented in rotavirus infections; however, the clinical implications remain unclear [2].

*Shigella* exhibits relative resistance to stomach acid, requiring only a few cells to induce disease. It replicates in the small bowel after ingestion and before reaching the colon, where it generates Shigella enterotoxins and serotype toxin 1, resulting in either watery or bloody diarrhea. Clinical presentations including elevated fever, nausea, and widespread colicky abdominal discomfort, followed by bloody mucoid diarrhea and tenesmus, which usually manifests between 12 h and 3 days after bacterial intake.

Breastfeeding serves as a preventive measure against gastrointestinal diseases in infants, as a significant portion of the immunoglobulins in human milk, primarily IgA, protects against enteric infections like rotavirus [6]. Immunoglobulins can be detected in the stools of breast-fed infants but not in those formula-fed [7]. However, the decline in breastfeeding due to the lifestyle in developed countries necessitates alternative preventive strategies. Although various rotavirus vaccines have been developed and proven effective in protecting against rotavirus [8,9,10], their efficacy is not 100% in all cases, prompting the need for complementary preventive approaches to reduce morbidity and mortality associated with rotavirus diarrhea [11].

In recent years, scientists have directed their efforts toward exploring the possible use of probiotic bacteria to combat the pathogens responsible for this gastroenteritis. Administering probiotic microorganisms is known to prevent several infections, allergic disorders, diarrhea, and inflammatory diseases, including inflammatory bowel disease (IBD) [12].

*Bifidobacterium longum* subsp. *infantis* stands out as one of the prevailing bacteria in the intestinal microbiota of infants solely breastfed, commonly found in the breast milk consumed by the infant. The research reported within this review provides preclinical and clinical evidence of the potential effects the probiotic *B. longum* subsp. *infantis* IM1^®^ can have in the management of both bacterial and viral gastrointestinal infections that cause diarrhea [13].

## 2. Relevant Literature

### 2.1. Isolation, Identification, and Phenotypic and Genotypic Characterization of B. longum subsp. infantis IM1^®^

Moreno et al., 2011 [13] described how *B. infantis* IM1^®^ was isolated from stool of an exclusively breastfed infant in the presence of gastrointestinal juices and physiological concentrations of bile salts. The authors carried out a study of the carbohydrate utilization profile using API 50CH (bioMérieux España S.A, Madrid, Spain) and of undesirable metabolic activities using API ZYM (bioMérieux España S.A), confirming that this probiotic does not have any unwanted enzymatic activity and has a great capacity for carbohydrate utilization, something already described in the scientific literature for the genus *Bifidobacterium* [13].

Identification of the probiotic strain was performed by sequencing the 16S ribosomal gene [13]. The complete genomic sequence of *B. infantis* IM1^®^ was obtained using PacBio’s single-molecule real-time DNA sequencing (SMRT) technology; the genome was circularized with an estimated size of 2,455,085 nt. No plasmids were detected, allowing an analysis of the functional activity of the probiotic genome and its definitive genotypic characterization. No mobile genetic markers of antibiotic resistance were detected in the genome of the bacteria nor were any genes of virulence or deleterious enzymatic activities [14].

The resistance to gastrointestinal juices, biliary salts, NaCl, and low pH of *B. infantis* IM1^®^ was evaluated by Moreno et al., 2011 [13], showing good abilities to resist high concentrations of bile salts and NaCl, as well as low pH, and therefore showing good survival abilities when passing through the gastrointestinal tract. In the same paper, the authors report sensitivity to antibiotics in terms of MICs, the capability to adhere to intestinal mucus, and the ability to produce lactic acid, deconjugate biliary salts, and for biogenic amines (histamine, cadaverine, tyramine, and putrescine). The results obtained demonstrated that *B. infantis* IM1^®^ was not resistant to any of the antibiotics tested; it had very good adhesion to the intestinal mucus (even higher than those of *Lacticaseibacillus rhamnosus* LGG); it did not deconjugate bile salts; and its ability to producing biogenic amines was negligible.

### 2.2. Pre-Clinical Studies in Cell Lines

Moreno et al., 2011 [13] carried out rotavirus propagation and in vitro inhibition assays as well as in vitro competition assays of *B. infantis* IM1^®^ against cellular binding sites. Studies were carried out on the ability of *B. infantis* IM1^®^ to inhibit the replication of Wa rotavirus (Strategy A) and the ability to protect cells from infection by the virus in MA-104 and HT-29 cell lines (Strategy B). These in vitro studies in MA-104 and HT-29 cells (Table 1) showed that *B. infantis* IM1^®^ can inhibit the replication of Wa rotavirus (up to 36.05% reduction in infectious foci) and protect epithelial cells from virus infection (up to 48.50% reduction in infectious foci).

Ruiz et al., 2020 [15] carried out an evaluation of the ability of *B. infantis* IM1^®^ to remove and reduce the binding of different bacterial enteropathogens (*Escherichia coli* LMG2092, *Clostridium dificile* LMG21717, *Cronobacter sakazakii* LMG5740, *Listeria monocytogenes* LMG13305, *Salmonella enterica* subsp. *enterica* LMG15860, *Shigella sonnei* LMG10473, and *Yersinia enterocolitica* LMG7889) to HT29 cells in the presence of galactooligosaccharides (GOS) to HT-29 and MA-104 cell lines. These authors evaluated the ability of *B. infantis* IM1^®^ to use different oligosaccharides or mixtures of oligosaccharides, allowing the identification of galactooligosaccharides (GOS) and mixtures containing GOS as the ones that increased the growth of *B. infantis* IM1^®^ the most. In the same publication, the symbiotic combination of GOS together with *B. infantis* IM1^®^ was used to examine the antimicrobial activity in coculture experiments against *E. coli*, *S. enterica*, *C. sakazakii*, *S. sonnei*, *L. monocytogenes*, and *C. difficile*. The symbiotic combination inhibited the growth of *C. difficile* but failed to inhibit *E. coli*, while *B. infantis* IM1^®^ alone without the help of GOS managed to reduce the growth of *C. sakazakii* by 1–2 log values during coculture with *B. infantis* IM1^®^. Similar results were reported previously [16] with other probiotic species against *S. typhimurium*. This aligns with earlier studies that illustrated the inhibition of *C. difficile* by culture supernatants of *B. longum* and *Bifidobacterium breve* strains. The inhibition effect was found to be dependent on the prebiotic substrates used for bifidobacteria growth [17]. Moreover, it aligns with certain studies proposing that *Bifidobacterium* strains could serve as an effective treatment for *Clostridium*-associated diarrhea [18,19]. Nevertheless, the exact mechanisms accountable for impeding pathogen growth in co-cultures with *Bifidobacterium* strains remain not entirely understood. The potential influence of pH reductions or the generation of antimicrobial substances effective against pathogenic bacteria, as illustrated for other species/strains [20], cannot be ruled out and merits additional exploration.

In the same publication by Ruiz et al., 2020 [15], experiments were conducted to assess *B. infantis* IM1^®^ pathogen displacement and the prevention of pathogen adhesion to enterocytes using the intestinal cell line HT29. *B. animalis* subsp. *lactis* Bb12 was employed as a probiotic control for comparative purposes. In these experiments, *B. infantis* IM1^®^ successfully displaced all the tested pathogenic strains (*C. sakazakii*, *E. coli*, *S. enterica*, *S. sonnei*, and *Y. enterocolitica*) that were previously attached to HT-29 cells. The displacement of *C. sakazakii* and *S. enterica* occurred similarly to that of the *B. animalis* subsp. *lactis* Bb12 strain. Moreover, the prior adhesion of *B. infantis* IM1^®^ to HT-29 cells reduced the adhesion of all pathogens to cells, with a more pronounced effect observed for *S. sonnei* and *C. sakazakii*. Overall, from the published results, it appears that *B. infantis* IM1^®^ was more efficient in displacing pathogens than the control probiotic (strain Bb12).

### 2.3. Purification and Identification of the Active Substance against Rotavirus Produced by B. infantis IM1^®^

Chenoll et al., 2016 [16] carried out studies to pinpoint the active peptide against rotavirus, which began with its isolation from the supernatant of *B. infantis* IM1^®^. Through protease digestions of the supernatant of *B. infantis* IM1^®^, both the protein nature of the active ingredient and the fact that the molecule responsible for inhibiting rotavirus replication was released into the supernatant were revealed. Reverse phase fractions were selected for rotavirus inhibition assays. All fractions that inhibited infection in rotavirus cell culture were analyzed by MALDI-TOF to obtain their peptide fingerprint and determine the molecular weights of the peptides identified in each fraction. The authors were able to identify the functional compound produced by *B. infantis* IM1^®^ responsible for hindering rotavirus replication both in vitro and in vivo. The active molecule was identified as an 11-amino-acid peptide (MHQPHQPLPPT, referred to as the 11-mer peptide) with a molecular weight of 1.282 KDa. The activity of the 11-mer peptide was confirmed by employing the synthesized peptide in Wa, Ito, and VA70 rotavirus infections of the HT-29 and MA-104 cell lines. Finally, protease activity was detected in the supernatant of *B. infantis* IM1^®^, responsible for the release of the 11-mer peptide. In this same study, the protease involved was preliminarily identified.

### 2.4. Studies in Animal Models with B. infantis against Pathogens That Cause Diarrhea

#### 2.4.1. Study of the Antirotaviral Capacity of *B. infantis* IM1^®^ In Vivo in a Mouse Model

Moreno et al., 2011 [13] carried out studies of the antirotaviral capacity of *B. infantis* IM1^®^ in an in vivo mouse model, using two sets of nine 8-week-old BALB/c. Each mouse in the reference group was administered 100 µL of sodium bicarbonate buffer (0.2 M), whereas the intervention group was subjected to an oral gavage of 10^9^ CFU of *B. infantis* IM1^®^ suspended in 100 µL of the identical buffer. After the first week, each mouse in the experimental group was given a probiotic booster dose repeatedly for the next 4 days. Subsequently, all mice in both groups were inoculated by oral gavage with 100 DD50 of murine McN virus in 50 µL of Earle’s balanced salt solution, and the feces of each animal were collected daily for the next 9 days. The presence of *Bifidobacterium* and the progression of viral infection in mice were assessed using ELISA and determined by the quantity of rotavirus antigen excreted in the feces (Figure 1).

A significant initial retarding of the shedding of rotavirus was observed within the first 48 h after infection (10^6^ FFU/mL, compared to 4 × 10^7^ FFU/mL in the untreated control group [*p* < 0.01]). This delay was attributed to an initial reduction in viral replication levels in mice fed *B. infantis* IM1^®^. However, it is important to note that the virus eventually infected all groups of mice, as a high viral particle dose was administered. Notably, on day 7, the concentration of antigen was statistically lower in mice treated with the probiotic (*p* < 0.05). It was shown that the consumption of *B. infantis* IM1^®^ provided preliminary in vivo protection against McN murine rotavirus infection, producing a delay in infection and an acceleration in recovery from infection by accelerating the elimination of viral particles in animal feces [13].

#### 2.4.2. In Vivo Study of the Antibacterial Capacity of *B. infantis* IM1^®^ in Weaned Piglets

An oral challenge using a weaning piglet model of infection has been used in several experiments to study the capacity of the probiotic strain *B. infantis* IM1^®^ to prevent and fight the intestinal disease caused by different bacterial pathogens that cause diarrhea, specifically by *Salmonella enterica* serovar Typhimurium, an enterotoxigenic *E. coli* K88, or *E. coli* F4. Barba-Vidal et al., 2017 [17] observed a reduction in the presence of *Salmonella* in feces and less colonization of the ileum by *E. coli* K 88 (33% reduction for animals with countable coliforms) in animals that had consumed the probiotic IM1^®^. In another study with the same animal model, Barba-Vidal et al., 2017 [18] reported that consumption of *B. infantis* IM1^®^ in combinations with *B. animalis* subsp. *lactis* BPL6 produced a significant reduction in some symptoms of the infection caused by *Salmonella* in the animals that had consumed the probiotics, among which it is worth highlighting increased voluntary feed-intake, reduced fecal excretion of *Salmonella*, decreased rectal temperature, decreased diarrhea scores, improved fermentation profiles, a tendency to lower colonic ammonia concentrations, stimulation of the intestinal immune response by increasing villous intraepithelial lymphocytes, and improvements in the villous:crypt ratio.

Rodriguez-Sorrento et al., 2020 [19], using the weaning piglet model of infection, evaluated the efficacy of two probiotic strains (*B. infantis* IM1^®^ and *Lacticaseibacillus rhamnosus* HN001), combined or not with a prebiotic containing oligofructose-enriched inulin, against *Salmonella enterica* serovar Typhimurium. Regarding the experimental treatments, animals belonging to the PRO group experienced a faster clearance of the pathogen, with more pigs being negative for its excretion at the end of the study and recovering the impaired ileal villi/crypt ratio more rapidly.

Rodriguez-Sorrento et al., 2021 [21] evaluated the potential of a symbiotic combination of multistrain probiotic (*B. infantis* IM1^®^ and *L. rhamnosus* HN001) with or without galacto-oligosaccharides against enterotoxigenic *Escherichia coli* (ETEC) F4 infection in postweaning pigs. The authors reported reduced pig major acute-phase protein (Pig-MAP) levels on day 4 PI in the animal group consuming the synbiotic.

Rodriguez-Sorrento et al., 2022 [19] evaluated the potential of a synbiotic combination of the probiotic strain *B. infantis* IM1^®^ with oligofructose-enriched inulin against *Salmonella enterica* serovar Typhimurium in a weaning pigs model. The authors reported that the administration of the synbiotic increased intraepithelial lymphocytes in the group consuming the synbiotic.

### 2.5. Clinical Studies

Escribano et al., 2018 [22] conducted a multicenter, double-blind, randomized, controlled clinical trial to assess whether an infant formula enriched with the probiotic *B. infantis* IM1^®^ was effective in reducing the incidence of diarrhea in healthy full-term infants (<3 months). A total of 93 (probiotic) and 97 (control) infants were randomly assigned, and 73 (probiotic) and 78 (control) completed the 12-week follow-up. The evaluation encompassed episodes of diarrhea, growth, digestive symptoms, and the presence of bifidobacteria in feces. Over the entire study period, the median number of diarrhea events per child was 0.29 ± 1.07 in the control group and 0.05 ± 0.28 in the probiotic group (*p* = 0.059). This trend toward fewer diarrhea episodes in the probiotic group reached statistical significance at 8 weeks (Figure 2, 0.12 ± 0.47 vs. 0.0 ± 0.0 events/infant, *p* = 0.047). After 4 weeks, the incidence of constipation was higher (odds ratio [OR] 2.67 (1.09–6.50)), and the frequency of bowel movements was lower (2.0 ± 1.0 vs. 2.6 ± 1.3 stools/day, *p* = 0.038) in the control group. No differences were observed at other time points or in other digestive symptoms, growth, or formula intake.

No differences in the overall *Bifidobacterium* levels in the fecal samples among the investigated groups were observed by real-time PCR, although a significant increase in *B. infantis* IM1^®^ counts was observed in the probiotic group. This distinction became apparent after a 4-week intervention and persisted throughout the entire study period. This result was also reflected in the microbiota analysis, showing a rise in *B. longum* at the study’s conclusion in the probiotic group (*p* = 0.023), a trend not observed in the control group (*p* = 0.45).

Microbiota analysis revealed a decrease in the prevalence of pathogens (*Escherichia*, *Clostridium*, *Salmonella*, *Campylobacter*, and *Yersinia*) in the probiotic group at the 12-week mark compared to those using the control formula, which maintained values similar to those at baseline, but this decrease was statistically nonsignificant. Additionally, the control group’s significant increase in *Escherichia coli* at the final time point compared to that at baseline was notable, while the presence of this pathogen remained unchanged throughout the follow-up in the probiotic group.

As a summary, Figure 3 shows the studies carried out with *B. infantis* IM1^®^ analyzed in this review.

## 3. Discussion

A new strain of *Bifidobacterium longum* subsp. *infantis* was isolated from the stool from a baby exclusively fed with breast milk. The strain was deposited in the Spanish Type Culture Collection (CECT) as *B. longum* subsp. *infantis* CECT 7210 and commercially designated as *B. infantis* IM1^®^. *B. longum* subsp. *infantis* is considered one of the predominant bacteria in the intestinal microbiota of infants exclusively fed with breast milk and commonly present in the breast milk ingested by the baby from their mother. This new strain has the key characteristics to be considered a probiotic, such as resistance to gastrointestinal juices, resistance to low pH, bile salts, and high NaCl concentrations, as well as an excellent adherence to intestinal epithelial cells and no resistance to antibiotics. In terms of food safety, *B. infantis* IM1^®^ is considered safe since it does not produce undesirable metabolites, and no translocation of the bacteria to organs has been detected in acute ingestion studies conducted in immunosuppressed mice [13]. Obtaining the complete sequence of its genome has made it possible to identify it genotypically and rule out the presence of antibiotic resistance markers that could be transferred, as well as to identify genetic markers of virulence or undesirable metabolic activities [14]. The potentially probiotic properties described above have been collected and detailed in the scientific literature, establishing a scientific consensus on the matter [23].

The good adhesion capacity shown by IM1^®^ in the mucus adhesion experiments carried out by Moreno et al., 2011 [13] indicated the potential ability to block the binding of pathogens or their displacement in the gastrointestinal epithelium, something that was confirmed by the results obtained by Ruiz et al., 2020 [15]. They observed that IM1^®^ has the ability to block and displace the binding of different gastrointestinal pathogens in in vitro studies carried out with intestinal epithelial cell lines. The capacity of probiotics to adhere to the host does not necessarily guarantee a health benefit, but the attachment of probiotic bacteria to the intestinal host cell’s binding sites through competition can potentially play a protective role against enteropathogens [24]. Tytgat et al., (2016) [25] described a mechanism of the competitive exclusion of *Enterococcus faecium* by *L. rhamnosus* GG. In this study, the researchers demonstrated how the mucus-binding SpaCBA pili of LGG, SpaC proteins, or antibodies against SpaC decrease the adhesion of *E. faecium*, which has comparable pili structures. Although similar structures have been observed in *Bifidobacterium*, their role in the competitive exclusion of pathogens has not yet been elucidated [26]. The adhesion capacity of beneficial bacteria may also prolong their residence time in the gastrointestinal tract, extending their intended positive effects. For instance, this could enhance the local impact of probiotic-generated metabolites (such as short-chain fatty acids, SCFAs) or amplify the immunomodulatory effects by bacterial surface-expressed molecules functioning as ligands for host receptors in the intestinal epithelium, initiating signaling pathways [27,28,29,30]. Probiotic bacteria can engage with pattern recognition receptors (PRRs), including the Toll-like receptors (TLRs) found on dendritic cells and macrophages, through the microbe-associated molecular patterns (MAMPs) present in the bacterial cell surface or released into the environment. The intimate contact between beneficial bacteria and host immune cells could facilitate the interaction of surface-bound components and other molecules released by probiotics, setting off a signaling cascade that leads to immunomodulation. In some beneficial bacteria, MAMPs can involve filamentous structures on their surface (for example, pilis), as previously mentioned. Filamentous structures are external components of bacteria crucial for their adhesive capability and TLR-2 signaling [31,32].

In addition to the possibility that IM1^®^ can block the binding of rotavirus to the gastrointestinal epithelium, blocking or delaying rotavirus infection and, by extension, accelerating the elimination of the virus in the gastrointestinal tract due to the excellent adhesion properties to the epithelium, this bacteria has the ability to produce a peptide with antirotaviral activity. Chenoll et al., 2016 [16] identified the possibility of a second mechanism of action by this strain to inhibit infections by this viral pathogen.

Several proteins and peptides in probiotic bacteria with antiviral activity have been reported [33,34]. In *L. gasseri* SBT2055, the protein SRCAP, with activity against respiratory syncytial virus (RSV), was identified, although the exact mechanism by which SRCAP protein inhibits RSV replication requires additional research. *L. gasseri* SBT2055 also has the capability to protect mice against influenza virus (IFV) infection by enhancing the expression of the Mx1 and Oas1a genes, which are crucial for reducing the virus titer in the lungs [35]. Upregulated ISGs can inhibit IFV infection effectively. The proteins synthesized by probiotic strains (for example, *L. casei* and *B. adolescentis*) directly engage with viral surface glycoproteins (VP4 or VP7), preventing the virus from entering MA104 cells and/or adhering to the cells [36]. Peptides derived from probiotics may impede endocytosis by interfering with clathrin-coated pit formation, a process vital for virus entry into endosomes. This interference consequently inhibits the formation of the viral transcriptional complex. A study showed that the oligopeptides from probiotic bacterial and viral capsids can be conjugated, disrupting the lipid membrane. This disruption, occurring at specific concentrations, results in pore formation, enabling the diffusion of viral components from the viral cells [37]. The genome sequencing of various *Bifidobacterium* and *Lactobacillus* strains revealed the existence of elements like surface layer glycoprotein, which was associated with the cell envelope. This finding established a link between the strains’ abilities to bind to host cells and their capacity to neutralize viral effects, irrespective of the impact of their metabolites. These proteins play a crucial role in signaling dendritic cells (DCs) and the functioning of T cells [38]. Several possible mechanisms have been proposed to explain the partial protection versus human rotavirus infection induced by probiotic bacteria: (a) directly affecting the viral infection and (b) killing the virus by modifying the immune system of the host. In addition to *Bifidobacterium*, the existence of other probiotic bacteria with activity against rotavirus has been reported. Research conducted by Kandasamy et al., 2016 [39,40] showed how antiviral activity against rotavirus infection with *Escherichia coli* Nissle and *Lacticaseibacillus rhamnosus* (Strain GG) is mediated by modulating B-cell responses, also showing that they are therapeutically effective against rotavirus diarrhea through the TLR3 signaling pathway. Another probiotic that shows in vitro activity against rotavirus is *Bacillus Clausii*. Other studies made with *L. delbrueckii* OLL1073R-1 (LDR-1) showed how its EPS could improve the intestinal innate antiviral response and prevent intestinal viruses such as rotavirus [41,42,43].

The results obtained with *B. infantis* IM1^®^ using the weaning piglet model of infection showed how this probiotic strain manifests notable positive activity against the gastrointestinal infections caused by the bacterial pathogens that cause diarrhea in an animal model that was not murine. These positive effects included a reduction in the symptoms of infections caused by *E. coli* and *S. enterica* (reduced fecal excretion of bacterial pathogen, decreased rectal temperature, decreased diarrhea scores), a faster recovery in infected animals (improvements in the villous:crypt ratio, reduced pig major acute-phase protein), as well stimulation of the intestinal immune response by increasing villous intraepithelial lymphocytes that had consumed this probiotic.

The accumulated evidence obtained through in vitro studies in cell lines and in vivo studies with different animal models showed the potential use of the IM1^®^ strain to combat the infections that cause diarrhea, whether of bacterial or viral origin, showing activity against the main agents causing diarrhea in babies. This finding was confirmed in the results obtained in the clinical study carried out by Escribano et al., [22], where a reduction in diarrhea episodes was observed in a group of babies who consumed formula with the IM1^®^ probiotic. They also observed a significant decrease in the presence of bacterial pathogens such as *E. coli* in the feces of these babies.

The latest position paper from the ESPGHAN Special Interest Group on Gut Microbiota and Modifications, released in 2023, issues weak recommendations for the utilization of probiotics, specifically *S. boulardii* and *L. rhamnosus*, in the management of acute pediatric gastroenteritis. These recommendations are grounded on weak evidence [44]. A lower level of evidence supports the use of *L. reuteri* for the same purpose, while a recommendation advises against the use of *Bacillus clausii,* as well as the combination of *L. reuteri* and *L. helveticus* in childhood diarrhea, in alignment with their preceding publications [45,46]. The authors refrained from offering specific guidance for diarrhea with a confirmed etiology and recognized the potential impact of rotavirus vaccine coverage on their examination of previously published data [45,47]. Additionally, *L. reuteri* was proposed as a potential preventive agent against diarrhea in preschool children, as it aids in preserving the integrity of the intestinal mucosa epithelial barrier, offering defense against the disruption commonly induced by enterotoxigenic *E. coli* infection [48]. Additionally, *L. plantarum* hinders the attachment of enteropathogenic *E. coli* to intestinal epithelial cells [49]. However, findings from in vitro studies still require validation through extensive cohort investigations. As recommended by Depoorter et al., [50] in their analysis, there is generally insufficient proof to routinely endorse probiotics for the prevention of acute gastroenteritis in children.

Consequently, the latest Cochrane systematic review underscores that there is an insufficiency of adequate data to facilitate a pathogen-specific examination of the role of probiotic supplementation in acute gastroenteritis. This constraint stems from the paucity of studies that have identified the etiological agents of diarrhea or have carried out a distinct statistical assessment specifically concentrating on a particular pathogen. Furthermore, the issue of achieving a balanced dosage ensuring both safety and efficacy for even the most utilized strains remains unresolved. Authorities in the domain propose conducting trials that compare the administration of diverse dosages of the same or various strains [51,52].

The studies described in this review are summarized in Table 2 and provide both preclinical and clinical evidence that other probiotic bacteria such as *Bifidobacterium longum* subsp. *infantis*, and particularly the *B. infantis* IM1^®^ strain, are emerging as probiotics that may be useful for the management of bacterial or viral gastroenteritis in babies. It is an opportunity for these probiotics to be included in this type of scientific consensus, increasing the therapeutic arsenal for their clinical use, improving the quality of evidence.

## 4. Conclusions

As a summary, the results showed that *B. infantis* IM1^®^ can be considered a probiotic capable of inhibiting rotavirus infection in in vitro and in vivo studies and the growth of gastrointestinal pathogens such as *S. tiphymurium*, *E. coli*, *C. difficile*, and *C. sakazakii*. *B. infantis* IM1^®^ is also able to displace some pathogens from the enterocyte layer, especially *C. sakazakii* and *S. enterica*, and prevent the adhesion of *C. sakazakii* and *S. sonnei*. In babies fed, an infant formula supplemented with *B. infantis* IM1^®^ could reduce episodes of diarrhea, being safe, well tolerated, and associated with a lower prevalence of constipation.

## 5. Future Directions

While the present review sheds light on the capacity of *Bifidobacterium longun* subsp. *infantis* IM1^®^ against the most frequent pathogen bacteria and viruses causing diarrhea in those of pediatric age, there remain several promising avenues for future exploration and development. The mechanism by which IM1^®^ is capable of displacing pathogens from the intestinal epithelium has not yet been elucidated, which is why it is one of the future lines of research. This knowledge could serve to improve this capacity in the probiotic, and therefore, additionally improve, if possible, its probiotic potential. Another interesting line of research for the future would be the integration of multiomics approaches. Integrating multiomics approaches, encompassing genomics, transcriptomics, proteomics, and metabolomics, holds great potential for unraveling the complex interactions among the probiotics, microbiota, and host during infectious processes. By combining data from multiple molecular layers, research can gain a more comprehensive understanding of the biological pathways and networks involved. More interventional clinical studies in babies with this probiotic strain would be necessary to reinforce the already existing results; moreover, the performance of longitudinal clinical studies would help to establish the long-term effects of the probiotic. These studies could provide valuable insights into the causal relationships between consumption of this probiotic bacterium and the reduction in the symptoms caused by gastrointestinal pathogens during infectious processes such as diarrhea and as well as the long-term effects of its consumption in this population.

## Figures and Tables

**Figure 1 microorganisms-12-01183-f001:**
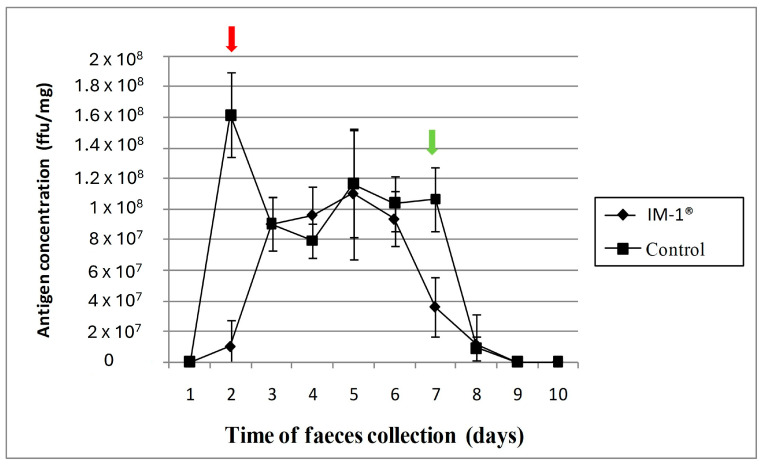
Rotavirus antigen excretion in the feces of mice administered probiotic *B. infantis* IM1^®^ was compared with that of the control group. The findings are presented as the mean value ± standard deviation of triplicate absorbance values at 492 nm, converted to FFU/mg using a standard curve. This curve shows the viral shedding for each mouse group, showing lower rotavirus antigen in IM1^®^ group on day 2 (red arrow) and day 7 (green arrow) compared with that in the control group (*p* < 0.01; *p* < 0.05, respectively).

**Figure 2 microorganisms-12-01183-f002:**
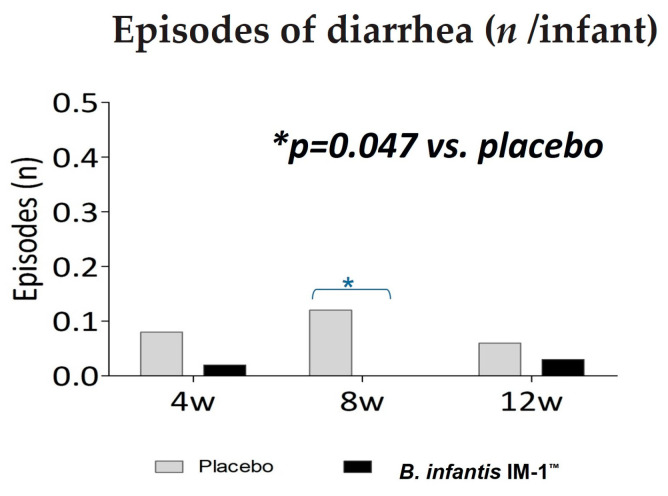
Number of diarrhea episodes over the study period by feeding group.

**Figure 3 microorganisms-12-01183-f003:**
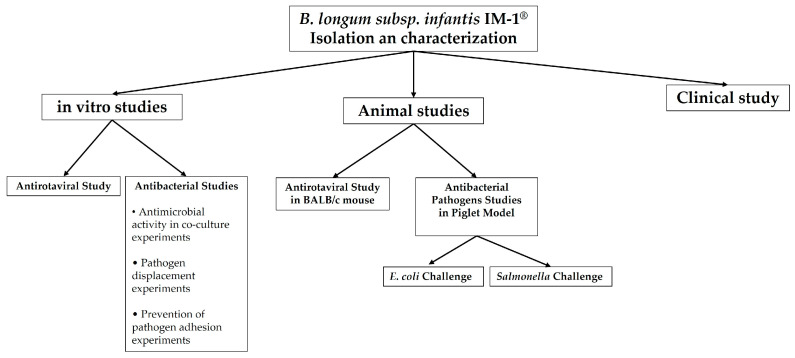
Global layout of the IM1^®^ studies reviewed [13,14,15,16,17,18,19,21,22].

**Table 1 microorganisms-12-01183-t001:** Activity against rotavirus Wa in HT-29 and MA-104 cell lines of different *B. infantis* strains using in vitro competition assays (strategies A and B) ^a^.

% Focus Reduction (Mean ± SD) in Cell Line				
Strain	HT-29		MA-104	
	Strategy A	Strategy B	Strategy A	Strategy B
*B. infantis* OR1	29.45 ± 1.20	45.20 ± 2.20	13.50 ± 2.16	24.80 ± 3.15
*B. infantis* OR2	26.15 ± 14.44	38.80 ± 1.75	15.00 ± 1.73	23.80 ± 2.11
*B. infantis* OR3	40.27 ± 2.45	40.00 ± 2.34	14.70 ± 3.54	24.40 ± 1.46
*B. infantis* IM1^®^	36.05 ± 4.23	48.50 ± 1.64	18.20 ± 4.41	31.80 ± 3.30
*B. infantis* OR5	49.83 ± 10.43	45.00 ± 3.89	9.00 ± 1.35	24.60 ± 4.25
*B. infantis* OR6	22.56 ± 15.20	39.40 ± 4.74	16.40 ± 1.66	30.20 ± 2.84

^a^ Results were obtained from three distinct experiments and are presented as the percentage reduction in infectious foci.

**Table 2 microorganisms-12-01183-t002:** Summary of the results obtained in in vitro experiments, animal models, and clinical studies against viral or bacterial pathogens causing diarrhea.

	Results	Reference
In Vitro Studies		
*Antirotaviral Study*	In MA-104 and HT-29 cells, *B. infantis* IM1^®^ demonstrates the ability to impede the replication of Wa rotavirus, resulting in a noteworthy 36.05% decrease in infectious foci. Additionally, it provides a safeguard for epithelial cells against virus infection, manifesting a substantial 48.50% reduction in infectious foci.	[13]
*Antibacterial Studies*		
Antimicrobial activity in co-culture experiments	When combined with GOS, *B. infantis* IM1^®^ successfully hindered the growth of *C. difficile*. Alone, without GOS assistance, *B. infantis* IM1^®^ achieved a reduction in the growth of *C. sakazakii* by 1–2 log counts during coculture.	[15]
Pathogen displacement experiments	The *B. infantis* IM1^®^ strain displaces pathogenic bacteria, including *Cronobacter sakazakii*, *Escherichia coli*, *Salmonella enterica*, *Shigella sonnei*, and *Yersinia enterocolitica*, which previously adhered to HT-29 cells. The displacement of *C. sakazakii* and *S. enterica* exhibited a similar result to that observed with the *B. animalis* subsp. *lactis* Bb12 strain utilized as a control.	[15]
Prevention of pathogen adhesion experiments	The binding of *B. infantis* IM1^®^ to HT-29 cells decreased the attachment of all pathogens (*Cronobacter sakazakii*, *Escherichia coli*, *Salmonella enterica*, *Shigella sonnei*, and *Yersinia enterocolitica*) to the cells, with a more noticeable impact observed for *S. sonnei* and *C. sakazakii*	[15]
**Animal Studies**		
*Antirotaviral Study in BALB/c Mice*	A notable initial delay in the shedding of rotavirus was noted within the initial 48 h after infection (10^6^ FFU/mL, compared to 4 × 10^7^ FFU/mL in the untreated control group [*p* < 0.01]). Delay could be ascribed to an initial decline in viral replication levels in mice that were administered *B. infantis* IM1^®^. By day 7 after infection, the antigen concentration was statistically lower in mice subjected to the probiotic treatment (*p* < 0.05).	[13]
*Antibacterial Pathogen Studies in Piglet Model*		
* E. coli* challenge	Diminishment ileal colonization (*p* = 0.077) Enhancement in the fermentation profile through elevated levels of butyric acid in nonchallenged piglets Augmentation of the villus:crypt ratio (*p* = 0.006)	[17,18]
* Salmonella* challenge	Diminishment pathogen excretion (*p* = 0.043) Augmentation intraepithelial lymphocytes (*p* = 0.002) Enhancement in the fermentation profile through elevated levels of butyric in piglets not subjected to challenges Augmentation ileal acetic acid (*p* = 0.008]) Augmentation villous:crypt ratio (*p* = 0.011) Reduced diarrhea scores in the probiotic group (*p* = 0.014) Reduced colonic ammonia concentrations [*p* = 0.078]	[19,21]
**Clinical Studies**	Fewer instances of diarrhea in the group of infants consuming the probiotic (*p* = 0.047). After 4 weeks in the control group (babies not consuming probiotic), there was an elevated occurrence of constipation and a decrease in stool frequency (*p* = 0.038).	[22]

## Data Availability

No new data were created or analyzed in this study. Data sharing is not applicable to this article.

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
