# Peer review of "A Probiotic Bacterium with Activity against the Most Frequent Bacteria and Viruses Causing Pediatric Diarrhea: Bifidobacterium longum subsp. infantis CECT 7210 (B. infantis IM1®)"

_microorganisms, 2024, doi:10.3390/microorganisms12061183_

Round 1

Reviewer 1 Report

Comments and Suggestions for Authors

Probiotics, which include bacteria and yeast, are live microorganisms that have beneficial effects on human health. Probiotics, particularly bifidobacteria, have been linked to a decrease in both the frequency and severity of diarrhoea episodes in infants. These effects include the prevention of colonization, cellular adhesion, and invasion by pathogenic organisms, direct antimicrobial activity, and modulation of the host immune response. Clinical trials have shown that probiotics can shape the intestinal microbiota, potentially controlling bowel diseases and promoting overall wellness.

The manuscript is properly structured, including all the necessary sections, such as Introduction, Materials and Methods, Results and Discussion and Conclusion.  The reference list consists of 52 sources, with 25% of them published between 2019 and 2023. Each section is clearly written and easily understandable. The text is written in a clear and concise manner. The manuscript offers valuable insight for experts in medical microbiology, biotechnology, and probiotics production. Overall, the manuscript has received positive feedback. However, there are some questions for the authors, as noted below.

1.                            During a multicentre, double-blind, randomised, controlled clinical trial, the number of diarrhoea episodes was studied over the study period for each feeding group. Did you examine the gut microbiota in addition to the symptoms of diarrhoea? Was there any change?

2.                            It is recommended to discuss how the strain of bifidobacteria being investigated may impact the composition of the gut microbiota.

Author Response

Questions Reviewer 1

We appreciate the review time dedicated to this article and below we answer the questions posed by the reviewer and indicate the changes made to the text of the paper draft following the reviewer's comments.

Question 1. During a multicentre, double-blind, randomised, controlled clinical trial, the number of diarrhoea episodes was studied over the study period for each feeding group. Did you examine the gut microbiota in addition to the symptoms of diarrhoea? Was there any change?

Answer 1: We monitored the microbiota in the babies throughout the study by collecting stool samples at baseline evaluation, at 4 weeks, at 8 weeks and at 12 weeks, unfortunately, the stool collection points were pre-established and did not depend on whether or not the baby had diarrhea at that time. Therefore, we cannot establish relationships between microbiota and diarrhea for the groups studied in this study.

Question 2. It is recommended to discuss how the strain of bifidobacteria being investigated may impact the composition of the gut microbiota.

Answer 2: We monitored the microbiota in the babies throughout the study by collecting stool samples at baseline evaluation, at 4 weeks, at 8 weeks and at 12 weeks, and analyzing the presence of Bifidobacterium and Bifidobacterium infantis IM-1® by Real Time PCR using specific primers for Bifidobacterium genus and Bifidobacterium infantis IM-1®, we also performed targeted sequencing of 16S RNA of all the bacteria present in the samples.

Real-time PCR results showed no differences in total Bifidobacterium levels in the feces between groups. However, as expected, there was a significant increase in B. infantis IM1 counts in the Probiotic group. These differences appeared after 4 weeks of intervention and were maintained throughout the study period. This effect was reproduced in the microbiota analysis, where an increase in B. longum was detected at the final point in the Probiotic group (P =0.023), that was not observed in the Control (P= 0.45).

Microbiota analysis pinpoints a nonsignificant decrease in the presence of pathogens (Escherichia, Clostridium, Salmonella, Campylobacter, and Yersinia) in the Probiotic group at 12 weeks compared with those taking the Control formula, which did show equal values than at baseline. We also observed in the Control group a significant increase in Escherichia coli at final time point with respect to baseline, whereas the presence of the pathogen did not change along the follow-up in the Probiotic group.

Changes made

Line 136-151 of Materials and Methods we have include a paragraph describing the methodology used with the Microbiota analysis in stool samples

Line 332-345 of Results and Discussion we have include the microbiota changes obtained and the impact of B. infantis IM-1® in microbiota.

Reviewer 2 Report

Comments and Suggestions for Authors

The authors reviewed preclinical and clinical evidence of the efficacy of B. longum subsp. infantis IM-1® against various gastrointestinal pathogens using in vitro models, animal models, and clinical studies. However, there are few references, whether in vitro, animal or clinical studies, and such conclusions are not reliable. If there are too little literature on this B. longum subsp. infantis IM-1®, it is not appropriate to conduct a literature review. Therefore, I suggest rejection.

Author Response

Questions Reviewer 2

We appreciate the review time dedicated to this article and although we do not share the reviewer's opinion that B. longum subsp. infantis IM-1® does not have enough publications to make a review. A review of the evidence available so far is necessary and useful to understand how this species can help, through its inclusion in infant formulas, to reduce the incidence or reduce the symptoms of diarrhoea, one of the most frequent pathologies during the pediatric age.

Reviewer 3 Report

Comments and Suggestions for Authors

The scientific article is devoted to the problem of importance for the prevention of gastrointestinal diseases caused by viruses and bacterial pathogens. The peer-reviewed article presents preclinical and clinical evidence of the effectiveness of B. longum probiotic strain against gastrointestinal diseases in infants.

The article is sound and contains interesting discussion. The authors have carefully reviewed the scientific data and results presented in publications in this field; the article provides a significant list of references. All the data obtained are clearly presented and also compared with the results of other studies. Overall, the article is highly relevant, scientifically sound, and advanced in this sector of microbiology.

For the successful publication of this article, I recommend making minor corrections:

Firstly, in the Materials and Methods section there are many references to techniques described by other authors, but the design of your experiment is not clearly presented. I recommend to present the overall design of the experiment, as well as a short description of each methodology used (how the cell culture, animal and patient studies were organised).

 Line 343-349 - In the conclusion, you describe the specific pathogens against which the probiotic strains you have isolated are effective (S. tiphymurium, E. coli, C. difficile, and C. sakazakii). However, in the Results and Discussion section, you describe research on the efficacy of a bifidobacterium strain against viruses, and there is no data on bacteria in this study. I recommend you revise the conclusion and add a summary of the results given specifically in this article.

Author Response

Questions Reviewer 3

We appreciate the review time dedicated to this article and below we answer the questions/coments posed by the reviewer and indicate the changes made to the text of the draft paper following the reviewer's comments.

Reviewer’s Coment 1: Line 343-349 (in the actual draft line 386-393) In the conclusion, you describe the specific pathogens against which the probiotic strains you have isolated are effective (S. tiphymurium, E. coli, C. difficile, and C. sakazakii). However, in the Results and Discussion section, you describe research on the efficacy of a bifidobacterium strain against viruses, and there is no data on bacteria in this study. I recommend you revise the conclusion and add

Answer Reviewer’s Coment 1:

Line 191-245 we present the results obtained with B. infantis IM-1® (Ruiz et al. 2020 [15]) against different bacterial pathogens that cause diarrhea in babies, including (S. tiphymurium, E. coli, C. difficile and C. sakazakii and Shigella sonnei and Listeria monocytogenes) this in vitro experiment show that B. infantis IM-1® has antimicrobial activity in co-culture experiments against C. difficile and C. sakazakii.

Also we present the results of pathogen displacement and prevention of pathogen adhesion to enterocytes using the intestinal cell line HT29 obtained with B. infantis IM-1® against S. tiphymurium, E. coli, C. difficile and C. sakazakii and Shigella sonnei, Yersinia enterocolitica and Listeria monocytogenes. B. infantis IM-1® strain successfully displaced and reduced the adhesion all tested pathogenic strains,

Line 311-314 we mentioned studies in porcine models of infection that have allowed us to corroborate the effectiveness of B. infantis IM-1®, in reducing the symptoms of infections caused by E. coli and S. enterica as well as accelerating recovery in infected animals that had consumed this probiotic [references 17-20 cited in the paper].

Changes made

Line 171 We have included a diagram (diagram 1) with the overall design of the experiments carried out in this paper and a short description of each methodology following the reviewer's recommendation. We hope that with this change the design of our experiments will be clearer to the reader.

Line 383 We have included a table as a summary of the results given specifically in this article following the reviewer's recomendation

Reviewer 4 Report

Comments and Suggestions for Authors

Good work, well done!.

I would like to have at the introduction and also into discussion a short comment about Bifidobacterium longum subsp. infantis as the principal bacteria in the intestinal microbiota of exclusively breastfed infants. 

Author Response

Questions Reviewer 4

We appreciate the time dedicated to reviewing the article and the reviewer's kind comments that encourage us to continue working and continue conducting more research, both preclinical and clinical, to delve deeper into the beneficial effects of B. infantis IM-1® in infant formulas. Following the reviewer's comments we have made the following changes to the paper draft:

Changes made

Line 70-72 we have include a short coment in introduction saying that “Bifidobacterium longum subsp. infantis is one of the most abundant bacteria in the intestinal microbiota of exclusively breastfed infants also being usually present in the breast milk that the baby ingests from their mother.”

Line 178-181 we have include a short coment in introduction saying that “Bifidobacterium longum subsp. infantis is one of the most abundant bacteria in the intestinal microbiota of exclusively breastfed infants also being usually present in the breast milk that the baby ingests.

Round 2

Reviewer 2 Report

Comments and Suggestions for Authors

I have no additional comments.

Author Response

We don´t have reviewer's comments in this Review Report 2

Reviewer 3 Report

Comments and Suggestions for Authors

The authors have done a significant revision of the manuscript, I suppose that the article can be published in the Microorganisms.

Reviewer 4 Report

Comments and Suggestions for Authors

I think the manuscript could be acceptable. Thank you for following the suggestions.